# Antibodies against SARS-CoV-2 among health care workers in a country with low burden of COVID-19

Mina Psichogiou[1]*, Andreas Karabinis[2], Ioanna D. Pavlopoulou[3], Dimitrios Basoulis[1], Konstantinos Petsios[4], Sotirios Roussos[5], Maria Pratikaki[6], Edison Jahaj[6], Konstantinos Protopapas[7], Konstantinos Leontis[8], Vasiliki Rapti[8], Anastasia Kotanidou[6], Anastasia Antoniadou[7], Garyphallia Poulakou[8], Dimitrios Paraskevis[5], Vana Sypsa[5], Angelos Hatzakis[5]

1 First Department of Internal Medicine, Laiko General Hospital, Medical School, National and Kapodistrian University of Athens, Athens, Greece, 2 Onassis Cardiac Surgery Center, Athens, Greece, 3 Pediatric Research Laboratory, Faculty of Nursing, National and Kapodistrian University of Athens, Athens, Greece, 4 Clinical Research Office, Onassis Cardiac Surgery Center, Athens, Greece, 5 Department of Hygiene, Epidemiology and Medical Statistics, Medical School, National and Kapodistrian University of Athens, Athens, Greece, 6 1st Department of Critical Care & Pulmonary Services, Medical School, National and Kapodistrian University of Athens, Evangelismos Hospital, Athens, Greece, 7 4th Department of Internal Medicine, Medical School, National and Kapodistrian University of Athens, Athens, Greece, 8 3rd Department of Internal Medicine, Sotiria General Hospital, Medical School, National and Kapodistrian University of Athens, Athens, Greece

* mpsichog@med.uoa.gr

**Data Availability Statement:** All relevant data are available at the Pergamos Institutional Repository of the National and Kapodistrian University of

# Abstract

## Introduction

Greece is a country with limited spread of SARS-CoV-2 and cumulative infection attack rate of 0.12% (95% CI 0.06–0.26). Health care workers (HCWs) are a well-recognized risk group for COVID-19. The study aimed to estimate the seroprevalence of antibodies to SARS-CoV-2 in a nosocomial setting and assess potential risk factors.

## Methods

HCWs from two hospitals participated in the study. Hospital-1 was a tertiary university affiliated center, involved in the care of COVID-19 patients while hospital-2 was a tertiary specialized cardiac surgery center not involved in the care of these patients. A validated, CE, rapid, IgM/IgG antibody point-of-care test was used. Comparative performance with a reference globally available assay was assessed.

## Results

1,495 individuals consented to participate (response rate 77%). The anti-SARS-CoV-2 weighted prevalence was 1.26% (95% CI 0.43, 3.26) overall and 0.53% (95% CI 0.06, 2.78) and 2.70% (95% CI 0.57, 9.19) in hospital-1 and hospital-2, respectively although the study was underpowered to detect statistically significant differences. The overall, hospital-1, and hospital-2 seroprevalence was 10, 4 and 22 times higher than the estimated infection attack rate in general population, respectively. Suboptimal use of personal protective equipment was noted in both hospitals.

Athens, Greece at https://pergamos.lib.uoa.gr/uoa/dl/object/2928810.

**Funding:** Gilead Hellas (https://www.gilead.com/utility/global-operations/europe/greece/greek) has funded the study with a grant bearing an approval code number GR-515-03.2020 after the approval by the Gilead Hellas Grants Committee at the colloquy of March 23d, 2020. Recipient of the grant was Hellenic Scientific Society for the Study of AIDS and Sexually Transmitted Diseases. AH was the Principal Investigator. The funders had no role in the study design, data collection and analysis, decision to publish, or preparation of the manuscript.

**Competing interests:** The study received funding from Gilead Sciences. There is a potential conflict of interest for the primary investigator, HA due to the funding received from Gilead. No other participant in the study has received funds from Gilead in any form or has been employed by Gilead. No participants have received funds for consultation services or donations or worked for any commercial company that would constitute a conflict of interest for the authors, including but not limited to Abbott and GenBody. The aforementioned competing interest does not alter our adherence to PLOS ONE policies on sharing data and materials.

## Conclusions

These data have implications for the preparedness of a second wave of COVID-19 epidemic, given the low burden of SARS-CoV-2 infection rate, in concordance with national projections.

## Introduction

Coronavirus disease 2019 (COVID-2019) caused by a novel coronavirus [severe acute respiratory syndrome coronavirus-2 (SARS-CoV-2)] emerged in Wuhan, China in December 2019 and spread worldwide in 212 countries and territories causing more than 5.8 million cases and 360,000 deaths within a period of 5 months [1].

In Greece, the first COVID-19 case was diagnosed on February 26. On March 23, a nationwide lockdown was enforced to reduce ongoing virus transmission as a response to this pandemic. As of May 30, there were 2,915 confirmed cases and 175 related deaths in Greece with a death rate of 16 per 1,000,000 population, which is one of the lowest in Europe [2]. Modelling data suggest that by the end of April 2020, when the first wave of epidemic was completed, the infection attack rate in Greece was 0.12% (95% CrI: 0.06–0.26) which corresponds to 13,200 total infections (95% CrI: 6,206–27,700) and a case ascertainment rate of 19.1% (95% CI 9.1–40.6) [3].

Health care workers (HCWs) is a well-known risk group for coronavirus infections [4, 5], accounted for a significant proportion of COVID-19 infections worldwide. By February 24, 2020, 3,387 HCWs out of 77,262 (4.4%) cases reported in China were HCWs [6]. The majority of these HCWs were documented at Hubei province, the epicentre of the epidemic. In a comprehensive analysis of 9,684 HCWs from Tongji Hospital in Wuhan, Hubei, the symptomatic infection rate was 1.1% while the respective asymptomatic infection rate was estimated at 0.9%. Nurses held a higher infection risk than physicians [OR: 2.07 (95% CI 1.7–4.3)] [7]. The city of Daegu, South Korea, had the first large outbreak of COVID-19 outside China. 121 HCWs were infected with infection rates 4.42 cases/1000 compared with 2.72 in the general population. Among HCWs, the infection rates were 2.37, 4.85, and 5.14 cases/1,000 among doctors, nurses, and nurse assistants, respectively [8]. In a large study among HCWs from the Netherlands, of whom 9,705 were hospital employees, a total of 1,353 (14%) reported fever and respiratory symptoms. Of those, 86 (6%) were infected with SARS-CoV-2, representing 1% of all HCWs employed [9]. Higher infection rates of SARS-CoV-2 by RT-PCR, ranging from 5–44%, were observed in HCWs from UK, Spain, Italy and US [10–17].

Serologic methods based on antibody testing (anti-SARS-CoV-2) could provide a more accurate estimate of epidemic size by detecting diagnosed and undiagnosed cases. Antibody methods rely on detection of IgM, IgG, IgA, or total antibodies by a variety of methods [18, 19].

The prevalence of the SARS-CoV-2 antibodies among HCWs was assessed in a number of studies from countries with high burden of SARS-CoV-2 infection where the reported anti-SARS-CoV-2 seroprevalence ranges from 1.6–45.3% [20–25]. Few studies used serological methods in the context of outbreak investigation [26, 27].

This study aimed to assess the seroprevalence of antibodies to SARS-CoV-2 in HCWs of two Greek hospitals during the current epidemic and identify potential risk factors for infection.

## Patients and methods

### Study design

This cross-sectional study recruited HCWs aged more than 18 years from two hospitals. The designated hospital-1 is a 500-bed tertiary, university-affiliated General Hospital providing care to COVID-19 patients. Hospital-2 with 134 beds is a Cardiac Surgery Center not involved in the care of COVID-19. The eligible personnel was in total 1,952, 1,120 in hospital-1, and 832 in hospital-2.

Two groups were investigated 1) first-line health care workers (FL-HCWs), defined as personnel whose activities involve contact with patients, and 2) second-line health care workers (SL-HCWs), such as office employees, technical personnel, cleaning personnel etc. Testing was offered at one specified location in each hospital for a period of 4 weeks, 13 April-14 May 2020, and 30 April– 15 May 2020 in hospital-1 and -2, respectively. Up to the date of testing, hospital-1 diagnosed or offered care in 38 individuals with COVID-19 while none was cared for COVID-19 in hospital -2. Written informed consent was obtained from all participants who participated in the study. Participants were interviewed using a structured questionnaire including demographics, education, position within hospital, exposure to COVID-19, use of personal protective equipment (PPE), and symptoms related to COVID-19. The data were directly recorded in a secure database. All participants were immediately informed on their test results, and they were offered a short posttest counseling session. The study was approved by the Laiko General Hospital Scientific and Ethics Review Board (protocol number: 291/02-04-2020) and the Onassis Cardiac Surgery Center Scientific and Ethics Review Board (protocol number: 681/06-04-2020).

### Validation of SARS-CoV-2 antibody testing

Testing was based on the GeneBody COVID-19 IgM/IgG detection, which is a lateral flow chromatographic immunoassay for the rapid and differential test of immunoglobulin M and immunoglobulin G against SARS-CoV-2. Serologic testing for SARS-CoV-2 antibodies was performed using capillary blood according to the manufacturer's instructions (Genebody Inc.). Testing is conducted in cassettes provided by the manufacturers including positive and negative control bands. Samples were concluded as reactive if the IgM or the IgG or both bands were positive using a colorimetric reader (Confiscope G20 analyser). According to the manufacturer, the detection limit of the assay is 1.84 s/co for IgM and 1.57 s/co for IgG. Samples with s/co between 1.0 and 1.84 for IgM and between 1.0 and 1.57 for IgG are considered a grey zone or weakly positive. All positive or weekly positive individuals were immediately retested as well as 1–2 weeks later. Concordant results were considered positive. Sixteen individuals with weekly positive samples were retested 1–2 weeks later, 13 were negative, 1 seroconverted to positive and 2 were unavailable. Positive individuals were immediately retested as well as 1–2 weeks later. The concordant results were considered positive.

The GeneBody assay was validated with serological panel from 116 symptomatic, positive by RT-PCR. COVID-19 patients (panel A) and with a second panel (panel B) including 250 samples collected during 2018 before SARS-CoV-2 pandemic

All samples of panel A were also tested with an FDA approved (EUA) CLIA assay (anti-SARS-CoV-2 IgG, Abbott Diagnostics). Samples with Index s/c $\geq$ 1.4 and < 1.4 were considered positive and negative respectively.

Both assays were based on nucleocapsid protein as antigen.

Manufacturers' clinical sensitivity estimates were 95% and 100% for GeneBody and Abbott respectively for samples collected >14 days from symptom onset. Manufacturer's clinical specificity estimates for GeneBody were 98% for IgM and 99% for IgG while for Abbott was 99.6%.

Concordance of capillary and venous sample was assessed in 15 samples of panel A tested in the same day

The seropositivity rates of GeneBody and Abbott from symptoms onset is shown in Table 1. The clinical sensitivity of both assays peaked at days 20–29 and ranged from 60.1–92.3% for GeneBody and 50.0–92.3% for Abbott. The overall clinical sensitivity was 74.1% (95% CI, 65.2–81.8%) for GeneBody and 81.9% (95% CI, 73.7–88.4%) for Abbott (McNemar's P = 0.108).

The seropositivity rate of GeneBody in panel B was 0 out of 250 with an estimated clinical specificity 100% (95% CI, 97.6% - 100.0%).

The concordance of antibody testing of capillary and venous samples with GeneBody in 15 patients from panel A tested the same day was 100%.

## Statistical analysis

To calculate the prevalence of antibodies to SARS-CoV-2, firstly, we calculated the unweighted proportions of positive tests and then we obtained the prevalence after weighting for the age distribution of the adult population (18–69 years old) in Athens Metropolitan area from the 2011 census. Secondly, we adjusted the weighted proportion for the sensitivity (74.1%) and specificity (100.0%) of the test, as assessed from the validation in the serological panels A and B, using the epiR package (R version 3.6.3, R Foundation for Statistical Computing, Vienna, Austria).

## Results

### Anti-SARS-CoV-2 prevalence among health care workers

A total of 1,495 HCWs consented to participate. The overall participation rate was 77% (81% and 71% in hospitals-1 and 2, respectively). Of 1,495 individuals tested, 69.7% were women, 61.7% were aged 35 to 54 years old, with a mean age (SD) of 46.4 (10.3) years. FL-HCWs accounted for 73.4% (1,097/1,495) of those tested (714/898 and 383/597 for the two hospitals respectively). Participants reporting direct exposure to known cases of COVID-19 during clinical work amounted to 11.5% (82/714), out of which 95% reported full use of PPEs. Subjects' characteristics are listed in Table.

A total of 15 individuals tested positive for anti-SARS-CoV-2, eleven of them for IgG only, three for IgM only and one for both IgM/IgG. After adjusting for age and test performance— assuming 74.1% sensitivity and 100% specificity the weighted seroprevalence for anti-SARS-CoV-2 in the total population was 1.26% (95% CI 0.43, 3.26). The weighed seroprevalence in hospital-1 was 0.53% (95% CI 0.06, 2.78) and in hospital-2 2.82% (95% CI 0.60, 9.62) (Table 2). The seroprevalence was 10, 4 times and 22 times higher in the overall hospital population, in hospital-1 and in hospital-2, respectively compared with the general population [0.12%, (95%

**Table 1. Seropositivity rates from symptoms onset of SARS-CoV-2 antibody using GeneBody and Abbott assays.**

| Time from symptom onset (Days) | Genbody Positive/N (%) | Abbott Positive/N (%) | P |
|---|---|---|---|
| 0–7 | 6/10 (60) | 5/10 (50) | 0.99 |
| 8–14 | 18/30 (60) | 22/30 (73.3) | 0.344 |
| 15–21 | 16/22 (72.7) | 19/22 (86.4) | 0.250 |
| 22–29 | 24/26 (92.3) | 24/26 (92.3) | 0.99 |
| 30–59 | 22/28 (78.6) | 25/28 (89.3) | 0.375 |

**McNemar's P = 0.108.**

**Table 2. Socio-demographic characteristics and weighted prevalence of anti-SARS CoV-2 of 1,495 participants in two hospitals in Athens.**

| Covariate | Population (N) | Anti-SARS-CoV-2 (+) | Weighted prevalence with 95% CI[a] |
|---|---|---|---|
| Overall | 1,495 | 15 | 1.26 (0.43, 3.26) |
| Hospital | | | |
| Hospital-1 | 906 | 4 | 0.53 (0.06, 2.78) |
| Hospital-2 | 589 | 11 | 2.82 (0.60, 9.62) |
| Gender | | | |
| Male | 453 | 5 | 1.60 (0.13, 6.92) |
| Female | 1,042 | 10 | 1.14 (0.36, 3.68) |
| Age (y) | | | |
| 18–34 | 231 | 1 | 0.58 (0.01, 3.22) |
| 35–54 | 922 | 8 | 1.17 (0.51, 2.30) |
| 55–70 | 342 | 6 | 2.37 (0.87, 5.10) |
| Country of birth | | | |
| Greece | 1,355 | 14 | 1.29 (0.42, 3.45) |
| Other | 140 | 1 | 0.79 (0.02, 17.02) |
| Marital status | | | |
| Married | 910 | 11 | 1.19 (0.41, 7.03) |
| Divorced / widowed | 134 | 0 | 0.00 (0.00, 4.75) |
| Single | 451 | 4 | 1.62 (0.09, 7.66) |
| Members of household | | | |
| 1 | 260 | 2 | 1.07 (0.03, 7.97) |
| 2 | 407 | 2 | 0.77 (0.02, 5.17) |
| 3 | 313 | 5 | 1.80 (0.33, 10.85) |
| 4 | 383 | 5 | 1.55 (0.24, 12.28) |
| 5+ | 132 | 1 | 0.65 (0.02, 19.87) |
| Highest completed level of education | | | |
| Master's degree/Doctorate | 416 | 2 | 0.46 (0.06, 5.82) |
| University or equivalent | 632 | 10 | 2.12 (0.55, 6.09) |
| Technical education or below | 447 | 3 | 0.66 (0.06, 9.68) |
| Job title | | | |
| Healthcare workers | 1,097 | 11 | 1.20 (0.34, 3.61) |
| Nonhealthcare workers | 398 | 4 | 1.04 (0.29, 8.66) |
| Symptoms[b] | | | |
| Any symptom | 150 | 3 | 2.38 (0.18, 16.18) |
| No symptom | 1,345 | 12 | 1.15 (0.33, 3.29) |

[a] Weighted prevalence for age and test performance.

[b] Among fever, cough, and shortness of breath.

Crl: 0.06–0.26)] [3] although the differences were not statistically significant. No significant associations were noted in the seroprevalence according to gender, country of birth, education, number of members in the household, FL-HCWs, SL-HCWs and use of PPE. Anti-SARS--CoV-2 prevalence was higher with increasing age, but the trend was not statistically significant (p = 0.10). The use of PPE was suboptimal in both hospitals. In hospital-1 and among the personnel treating COVID-19 the use of gloves, masks, glasses, gown was 96%, 99%, 56% and 63%, respectively. In hospital-2 the use of gloves and mask was reported in 99.7% and 100% while the use of glasses and gown occasionally (15%). Amongst seropositive individuals, only 9 reported full use of PPE.

Among all participants, 150 (10.1%) reported some symptoms indicative of COVID-19 in the previous 3 months; 82 reported fever, and 111 of them cough; 27 reported shortness of breath. Overall, 1,345 (89.9%) reported no symptoms. The prevalence of anti-SARS-CoV-2 was 2.38% (95% CI 0.18, 16.18) and 1.10% (95%CI 0.33, 3.29) in those who reported and those not reporting symptoms, respectively but the difference was not statistically significant.

## Discussion

We used a validated point-of-care antibody test. We estimated de-novo the clinical sensitivity, the clinical specificity and the comparative performance of the point-of-care test with a reference globally available assay [28]. The clinical specificity of GeneBody was found 100% (95 CI, 97.6–100.0%). The clinical sensitivity was 74.1% (95 CI, 62.5–81.8%) and did not differ statistically from Abbott reference assay 81.9% (95 CI, 73.7–88.4%)

In this survey of SARS-CoV-2 antibodies among hospital personnel, the overall seroprevalence was 1.26% (95% CI 0.43, 3.26). This seroprevalence rate is consistent with the low burden of COVID-19 in Greece. However, in the total hospital population and in that of hospital-2, it was 10 and 22 times higher, respectively, compared to the cumulative infection attack rate estimated by mathematical modeling for the general population in Greece [3] and a recently published population serological survey in Greece [29]. This is not surprising since the spread of SARS-CoV-2 is highly heterogeneous. In New York State the prevalence of anti-SARS-CoV-2 was found 14.0% with a range of 3.6–22.7% [30].

Due to the low burden of infection, the study is underpowered for pointing out risk factors. The difference in the prevalence between hospital-1 [0.51% (95% CI 0.06, 2.66)] and hospital-2 [2.70% (95% CI 0.57, 9.19)] is not significant. However, it is consistent with data suggesting that HCWs in hospitals involved in COVID-19 care could have a lower burden of infection than those not participating in COVID-19 care [7, 31]. This is probably due to the more extensive use of PPE, which is the main determinant for risk of SARS-CoV-2 infection in the health care environment [32]. In this study the use of PPE was suboptimal in both hospitals. Other reported risk factors are working in high- risk departments, long duty hours, practicing suboptimal hand hygiene [33]. Of the 42,600 HCWs caring for COVID-19 patients in the second half of the China epidemic, none was infected, suggesting that sufficient precautions and rigorous enforcement of PPE are the major determinants for eliminating COVID-19 infection [6].

A further challenge is whether SARS-CoV-2 infection can be truly attributed to hospital-acquired infections, especially in countries with a high burden of community infection [28]. In the study of Lai Y et al, contact with patients (59%), colleagues with infection (11%), and community acquired infection (13%) were the main routes of exposure among HCWs [7]. Contradicting results are noted in two large studies from Madrid and Birmingham. The anti-SARS-CoV-2 prevalence is higher in HCWs working in areas with exposure to COVID-19 (31–34%) compared with low-risk area (26%) and external workers (30%) in Madrid [24]. On the contrary in Birmingham study the anti-SARS-CoV-2 prevalence was higher among general medicine and housekeeping general personnel (30–35%) compared with intensive care and emergency medicine (13–15%) [21].

Several study limitations are noted: 1) The sensitivity of the currently existing antibody assays is not well knownbeyond 3 months from infection. However, the study took place at the end of the 1st epidemic wave and the time of testing was within the 3 months period from the 1st reported case in Greece (February 26, 2020). Studies have shown a steady prevalence decrease with time [18, 19]. 2) The study, due to the low anti-SARS-CoV-2 prevalence, is underpowered to detect risk factors.

In conclusion, the burden of SARS-CoV-2 infection among hospital personnel in Athens by the end of first wave of SARS-CoV-2 is low, consistent with the low burden of infection in the country. The use of PPE was suboptimal. These findings have implications for the pre-paredness of a second wave of COVID-19 such as to urgently increase the availability of PPE and expedite hospital preparedness for possible major increase in the demand of hospitals and ICU beds.

## Supporting information

**S1 Appendix. Evaluation of sensitivity and specificity of GeneBody COVID-19 IgM/IgG antibody panel.**
(DOCX)

## Acknowledgments

The authors would like to thank for their assistance in testing the tested population and the accumulation of data: Panayiotis Axaopoulos, Georgios Goumas, Evangelos Kokolesis, Michaella Alexandrou, Sofia Radi, Dimitra Siakali, Erica Alexandrou, Dimosthenis Theodosia-dis, Ilias Sinanidis, Charalambos Kazamiakis. We would also like to thank the Onassis C.S.C. staff members: George Stravopodis & Sofia Hatzianastasiou, and the Advisory board members in Laiko General Hospital: John Boletis, Nikolaos Sypsas, Michalis Samarkos, Ioannis Floros, Theoni Zougkou, Amalia Karapanou, Michalis Sambanis

## Author Contributions

**Conceptualization:** Mina Psichogiou, Andreas Karabinis, Ioanna D. Pavlopoulou, Angelos Hatzakis.

**Formal analysis:** Dimitrios Basoulis, Sotirios Roussos, Vana Sypsa.

**Funding acquisition:** Angelos Hatzakis.

**Investigation:** Mina Psichogiou, Andreas Karabinis, Ioanna D. Pavlopoulou, Dimitrios Basou-lis, Maria Pratikaki, Edison Jahaj, Konstantinos Protopapas, Konstantinos Leontis, Vasiliki Rapti.

**Methodology:** Angelos Hatzakis.

**Project administration:** Mina Psichogiou, Konstantinos Petsios, Anastasia Kotanidou, Ana-stasia Antoniadou, Garyphallia Poulakou, Angelos Hatzakis.

**Resources:** Konstantinos Petsios.

**Supervision:** Andreas Karabinis, Anastasia Kotanidou, Anastasia Antoniadou, Garyphallia Poulakou, Angelos Hatzakis.

**Writing – original draft:** Mina Psichogiou, Sotirios Roussos, Dimitrios Paraskevis, Vana Sypsa, Angelos Hatzakis.

**Writing – review & editing:** Mina Psichogiou, Andreas Karabinis, Ioanna D. Pavlopoulou, Dimitrios Basoulis, Konstantinos Petsios, Sotirios Roussos, Maria Pratikaki, Edison Jahaj, Konstantinos Protopapas, Konstantinos Leontis, Vasiliki Rapti, Anastasia Kotanidou, Ana-stasia Antoniadou, Garyphallia Poulakou, Dimitrios Paraskevis, Vana Sypsa, Angelos Hatzakis.

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
