## [Decision Letter · Decision Letter 0]

21 Sep 2020

PONE-D-20-20858

Antibodies against SARS-CoV-2 among health care workers in a country with low burden of COVID-19

PLOS ONE

Dear Dr. Psichogiou,

Thank you for submitting your manuscript to PLOS ONE. After careful consideration, we feel that it has merit but does not fully meet PLOS ONE’s publication criteria as it currently stands. Therefore, we invite you to submit a revised version of the manuscript that addresses the points raised during the review process.

I would like to apologise for the delay of the decision, but under the current circumstances it was very difficult to find reviewers that were available. 

We look forward to receiving your revised manuscript.

Kind regards,

Ronald Dijkman, PhD

Academic Editor

PLOS ONE

Additional Editor Comments:

Based on the available literature and deficits of certain early phase commercial serological assays, and reviewer comments, I would recommend that the authors would confirm their finding with an additional serological assay, in order for the revised version of their manuscript to be considered.

Journal Requirements:

"Gilead Hellas (https://www.gilead.com/utility/global-operations/europe/greece/greek) has funded the study with a grant bearing an approval code number GR-515-03.2020  after the approval by the Gilead Hellas Grants Committee at the colloquy of March 23d, 2020. Recipient of the grant was AH. The funders had no role in the study design, data collection and analysis, decision to publish, or preparation of the manuscript."

We note that you received funding from a commercial source: Gilead Sciences.

Reviewers' comments:

Reviewer's Responses to Questions

**Comments to the Author**

1. Is the manuscript technically sound, and do the data support the conclusions?

Reviewer #1: Yes

Reviewer #2: No

2. Has the statistical analysis been performed appropriately and rigorously? 

Reviewer #1: No

Reviewer #2: No

3. Have the authors made all data underlying the findings in their manuscript fully available?

Reviewer #1: Yes

Reviewer #2: Yes

4. Is the manuscript presented in an intelligible fashion and written in standard English?

Reviewer #1: Yes

Reviewer #2: Yes

5. Review Comments to the Author

Reviewer #1: The manuscript by Psichogiou describes a study on health care workers in Greece. A low level of immunity was found which mirrors the low rate on infection in Greece. The low prevalence of people with antibodies to SARS-CoV-2 cannot draw conclusions on risk factors, which is unfortunate. Still the low prevalence is worthwhile sharing with the scientific community.

Suggestions for improvement;

- line 111, please add the viral protein that is used as target in the GeneBody COVID-19 antibody detection test.

- line 140-141 and line 158 – 159: It is mentioned that seroprevalence is 10, 4 or 20 times higher than the general population. However, these numbers are probably not significant as the CI of the hospital tests is so wide. Please provide P-values, and mention whether there is a significant difference (or not).

Reviewer #2: Pichogiou et al. perform a serological testing for COVID-19 among health care workers in Greece, a country with low burden of COVID-19. First, the major issue is lack of detailed description for the commercial kit from Genebody Inc they used for antibody testing. what is the catalogue Number? Is it a quantitative or qualitative gold immunochromatography assay? It will be great if the authors use alternative assay to confirm the positive samples. Otherwise, it is hard to know the seropositive individuals identified from this assay were really positive or false positive. Also, the author mentioned that "capillary blood" was used for testing, instead of venous blood. From our practical experience, capillary blood might cause false positivity.

Secondly, some key details of socio-demographic characteristics of HCWs in two hospitals were missing. For examples, how many individuals in hospital-1 were front line HCW (FL-HCW)? How many of them directly exposed to COVID-19 patients without proper PPE? How many seropositive HCWs used PPE? Only identifying the seroprevalence among HCWs has limited implication for preparedness of a second wave of COVID-19.

6. PLOS authors have the option to publish the peer review history of their article (what does this mean?). If published, this will include your full peer review and any attached files.

Reviewer #1: No

Reviewer #2: No

---

## [Author Response · Author response to Decision Letter 0]

24 Oct 2020

To the Editor of PLOS ONE

Dear Dr. Dijkman,

Thank you for the valuable comments. To address all the concerns related to anti-SARS-CoV-2 rapid test assay we performed de-novo estimation of clinical sensitivity, clinical specificity, comparative efficacy and we used a strict quality control to accept positivity for capillary blood samples. 

The clinical sensitivity and the comparative performance of GeneBody with Abbott antibody test (SARS-CoV-2 IgG – Abbott, Chicago. IL, USA) was assessed in panel A. Panel A includes 116 serum or plasma samples from 116 COVID-19 patients with clinical disease and positive RT-PCR. Both assays are based on the nucleocapsid antigen. There was no statistical significance difference in clinical sensitivity of GeneBody and Abbott assays and their peak clinical sensitivity was similar (Table 1). 

We de-novo estimated the clinical specificity of GeneBody assay on panel B, including 250 serological samples collected before COVID-19 epidemic. The specificity was found 100% , suggesting that the GeneBody assay is appropriate for studies in low prevalence populations. 

Further quality controls were included. Positive capillary samples by GeneBody were tested 3 times. Two times at the date of testing and a third time 1-2 weeks later. All samples exceeding the cut-off recommended by manufacturer (1.84 s/co for IgM and 1.57 s/co for IgG) were found positive in all tests. The great majority of samples scored as weakly positive or grey zone (1.0 – 1.84 s/co for IgM and 1.0 – 1.57 s/co for IgG) were found negative when tested 1-2 weeks later (see Validation of SARS-CoV-2 antibody testing). 

To assess the concordance between capillary and venous blood we tested serum samples and capillary results tested in the same day. There was 100% concordance. Finally, we adjusted the estimated prevalence for the clinical sensitivity and specificity of GeneBody assay (see Statistical analysis). 

The choice of GeneBody test was based on the preliminary evaluation of a handful number of rapid tests taking advantage of our panel A and panel B. Our study was the first population study of anti-SARS-CoV-2 in Greece and later studies for anti-SARS-CoV-2 confirmed our findings. 

Therefore, we believe that our results are valid and describe adequately the prevalence of anti-SARS-CoV-2 in the health care workers population in Greece.

Our detailed response in Reviewers follows:

Reviewer #1 Comments- Suggestions:

 Suggestions for improvement

- line 111, please add the viral protein that is used as target in the GeneBody COVID-19 antibody detection test.

- line 140-141 and line 158 – 159: It is mentioned that seroprevalence is 10, 4 or 20 times higher than the general population. However, these numbers are probably not significant as the CI of the hospital tests is so wide. Please provide P-values, and mention whether there is a significant difference (or not).

Response to Reviewer #1

Information pertaining to viral protein used in the assays was added in line 134 of the non-tracking final manuscript: “Both assays were based on nucleocapsid protein as antigen.”

The statistical non-significance was included in lines 183 of the non-tracking final manuscript: “…although the differences were not statistically significant.”

Reviewer #2 Comments- Suggestions: 

Comment 1: Pichogiou et al. perform a serological testing for COVID-19 among health care workers in Greece, a country with low burden of COVID-19. First, the major issue is lack of detailed description for the commercial kit from Genebody Inc they used for antibody testing. what is the catalogue Number? Is it a quantitative or qualitative gold immunochromatography assay? It will be great if the authors use alternative assay to confirm the positive samples. Otherwise, it is hard to know the seropositive individuals identified from this assay were really positive or false positive. Also, the author mentioned that "capillary blood" was used for testing, instead of venous blood. From our practical experience, capillary blood might cause false positivity.

Response to Reviewer #2

Comment 1: The clinical sensitivity and the comparative performance with Abbott test was assessed in panel A. Panel A includes 116 samples from 116 COVID-19 patients with clinical disease and positive RT-PCR. Both assays are based on the nucleocapsid antigen. There was no statistical significance difference in clinical sensitivity of GeneBody and Abbott assays and their peak sensitivity was similar (Table 1). 

We de-novo estimated the clinical specificity of GeneBody assay which was on panel B, including 250 serological samples collected before COVID-19 epidemic. The specificity was found 100% , suggesting that the GeneBody assay is appropriate for studies in low prevalence populations. 

Further quality controls were included. Positive samples in capillary blood were tested 3 times. Two times at the date of testing and a third time 1-2 weeks later. All samples exceeding the cut-off recommended by manufacturer (1.84 s/co for IgM and 1.57 s/co for IgG) were found positive. The great majority of samples scored as grey zone were found negative when tested 1-2 weeks later. 

To assess the concordance between capillary and venous blood we tested serum samples and capillary results tested in the same day. There was 100% concordance. 

The GeneBody assay is a qualitative lateral flow immunoassay with catalogue number COVI040. The GeneBody Inc. is one of the leaders in rapid diagnostic testing pioneered Zika virus diagnostic tests. 

Reviewer #2 Comments- Suggestions: 

Comment 2: Secondly, some key details of socio-demographic characteristics of HCWs in two hospitals were missing. For examples, how many individuals in hospital-1 were front line HCW (FL-HCW)? How many of them directly exposed to COVID-19 patients without proper PPE? How many seropositive HCWs used PPE? Only identifying the seroprevalence among HCWs has limited implication for preparedness of a second wave of COVID-19.

Response to Reviewer #2

Comment 2: FL-HCW and PPE use amongst FL-HCWs directly exposed to SARS-COV-2 information has been added at line 166-168 of the manuscript “FL-HCWs accounted for 73.4% (1,097/1,495) of those tested (714/898 and 383/597 for the two hospitals respectively). Participants reporting direct exposure to known cases of COVID-19 during clinical work amounted to 11.5% (82/714), out of which 95% reported full use of PPEs.”

Regarding use of PPE amongst seropositive participants, we added line 190-191 “Amongst seropositive individuals, only 9 reported full use of PPE.”

---

## [Decision Letter · Decision Letter 1]

16 Nov 2020

Antibodies against SARS-CoV-2 among health care workers in a country with low burden of COVID-19

PONE-D-20-20858R1

Dear Dr. Psichogiou,

We’re pleased to inform you that your manuscript has been judged scientifically suitable for publication and will be formally accepted for publication once it meets all outstanding technical requirements.

Kind regards,

Ronald Dijkman, PhD

Academic Editor

PLOS ONE

Additional Editor Comments (optional):

Reviewers' comments:

Reviewer's Responses to Questions

**Comments to the Author**

1. If the authors have adequately addressed your comments raised in a previous round of review and you feel that this manuscript is now acceptable for publication, you may indicate that here to bypass the “Comments to the Author” section, enter your conflict of interest statement in the “Confidential to Editor” section, and submit your "Accept" recommendation.

Reviewer #1: All comments have been addressed

2. Is the manuscript technically sound, and do the data support the conclusions?

Reviewer #1: Yes

3. Has the statistical analysis been performed appropriately and rigorously? 

Reviewer #1: Yes

4. Have the authors made all data underlying the findings in their manuscript fully available?

Reviewer #1: Yes

5. Is the manuscript presented in an intelligible fashion and written in standard English?

Reviewer #1: Yes

6. Review Comments to the Author

Reviewer #1: (No Response)

7. PLOS authors have the option to publish the peer review history of their article (what does this mean?). If published, this will include your full peer review and any attached files.

Reviewer #1: No

---

## [Editor Report · Acceptance letter]

23 Nov 2020

PONE-D-20-20858R1 

Antibodies against SARS-CoV-2 among health care workers in a country with low burden of COVID-19 

Dear Dr. Psichogiou:

I'm pleased to inform you that your manuscript has been deemed suitable for publication in PLOS ONE. Congratulations! Your manuscript is now with our production department. 

Kind regards, 

on behalf of

Dr. Ronald Dijkman 

Academic Editor

PLOS ONE